# An Analysis of the External Validity of EEG Spectral Power in an Uncontrolled Outdoor Environment during Default and Complex Neurocognitive States

**DOI:** 10.3390/brainsci11030330

**Published:** 2021-03-05

**Authors:** Dalton J. Edwards, Logan T. Trujillo

**Affiliations:** 1Department of Neuroscience, School of Behavioral and Brain Sciences, The University of Texas at Dallas, Dallas, TX 75080-3021, USA; dalton.edwards@utdallas.edu; 2Department of Psychology, Texas State University, San Marcos, TX 78666, USA

**Keywords:** QEEG, mobile EEG, PASAT, resting state, external validity

## Abstract

Traditionally, quantitative electroencephalography (QEEG) studies collect data within controlled laboratory environments that limit the external validity of scientific conclusions. To probe these validity limits, we used a mobile EEG system to record electrophysiological signals from human participants while they were located within a controlled laboratory environment and an uncontrolled outdoor environment exhibiting several moderate background influences. Participants performed two tasks during these recordings, one engaging brain activity related to several complex cognitive functions (number sense, attention, memory, executive function) and the other engaging two default brain states. We computed EEG spectral power over three frequency bands (theta: 4–7 Hz, alpha: 8–13 Hz, low beta: 14–20 Hz) where EEG oscillatory activity is known to correlate with the neurocognitive states engaged by these tasks. Null hypothesis significance testing yielded significant EEG power effects typical of the neurocognitive states engaged by each task, but only a beta-band power difference between the two background recording environments during the default brain state. Bayesian analysis showed that the remaining environment null effects were unlikely to reflect measurement insensitivities. This overall pattern of results supports the external validity of laboratory EEG power findings for complex and default neurocognitive states engaged within moderately uncontrolled environments.

## 1. Introduction

Quantitative electroencephalography (QEEG) involves the complex numerical analysis of digitally recorded EEG signals that can provide significant insight into the functional relevance of bioelectric brain activity [1]. Traditionally, these analyses have been applied to EEG data collected within controlled laboratory environments, primarily due to the physical limitations of the recording equipment; traditional EEG systems require large amplifiers and computers that cannot be easily transported. Nevertheless, EEG data collected from controlled recording environments are beneficial for quantitative analysis because such data reflects well-defined experimental manipulations and clean measurement of the EEG signals. This yields strong internal validity among independent and dependent variables and robust performance of numerical analysis algorithms applied to the EEG signals.

However, a major drawback to recording EEG in controlled environments is that the external validity of any quantitative results is limited [2,3]. There are three main factors limiting the external validity of these results [4]. First, active behavior of participants in most laboratory studies is highly constrained because the physical effects of gross motor movements often degrades the signal quality of noninvasive brain imaging technologies. Second, most laboratory studies utilize simple static, regularized stimuli that are crude approximations to the dynamic, irregular stimulation found in naturalistic environments. Third, laboratory studies are generally free of environmental background influences (static or dynamic, regular or irregular) that can modify neurocognitive performance. Thus, it remains unclear if the QEEG findings produced by controlled laboratory studies generalize to the case of data collected within uncontrolled environments.

Fortunately, mobile EEG technologies have recently emerged that have opened up new possibilities for the measurement of EEG signals related to active behavior conducted inside or outside of the laboratory [2]. This technology involves small, battery-powered, wearable EEG amplifiers that can record the brain signals of participants while they naturalistically engage in task performance within a variety of interactive environments [5,6,7,8,9,10,11]. Research using this technology is still in its early stages, with key methodological, analytical, and interpretational hurdles yet to be resolved [2,12,13,14,15,16]. Nevertheless, mobile EEG technology has reached the point that it can now be used to study the external validity of laboratory QEEG findings, which is a goal of the present study.

In exploring the external validity limits of laboratory-based QEEG findings, it is useful to take an incremental approach in which the main factors affecting these limits (listed earlier) are investigated separately. In the present study, we examined the effect of uncontrolled background influences in the physical environment because manipulating this factor is relatively straightforward and little study has been given to it to date. Previous mobile EEG studies have mainly focused on neurocognitive performance within a single physical or virtual environment (e.g., laboratory, classroom, outdoor, virtual) [8,11,12,17,18,19,20,21,22,23,24,25,26]. To our knowledge, only two mobile EEG studies to date [27,28] have directly compared human neurocognitive performance between different physical environments with different physical characteristics and levels of dynamic irregularity. The first study [27] examined differences between a controlled indoor laboratory and an uncontrolled outdoor bicycle pathway. However, cognitive task performance in this study (auditory oddball detection) was also paired with a different physical activity in each environment (sitting indoors, bicycling outside). The second study [28] removed this confound by examining cognition during bicycling activity within a quiet park and a noisy roadway. In the present study, we also recorded mobile EEG from participants performing neurocognitive tasks during the same physical activity within two different physical environments—sitting in a controlled laboratory environment (closed space, minimal noise, static temperature and atmosphere) and a moderately uncontrolled outdoor environment (open space, background noise, weather changes; see Figure 1).

Another important factor to consider when testing the external validity of laboratory QEEG findings is the particular perceptual, cognitive, and motor functions that an individual performs during mobile EEG recording. Previous mobile EEG studies have focused on several functions (vigilant attention, perceptual novelty and stimulus significance, motor activity) as engaged by a variety of simple and complex tasks [8,11,12,17,18,19,20,21,22,23,24,25,26,27]. However, certain tasks might be expected to be more greatly influenced by the background environment than others, particularly tasks engaging executive functioning in complex, flexible combination with other cognitive functions [29]. In the present study, participants performed the paced auditory serial addition test (PASAT) [30,31], a mental task that engages several complex cognitive functions, including number sense, attention, working memory, and executive control [32,33,34]. However, in order to assess QEEG metrics reflecting cognitive engagement during the PASAT, a comparison condition is needed in which cognition is relatively disengaged. Here the control condition was EEG activity recorded while participants performed a simple resting state task [35,36,37,38], where an individual remains in an unstimulated state of wakeful rest in a manner that engages a default mode of brain activity [39]. This resting task was also of interest in and of itself as a metric of the effects of background environment on the endogenous neurocognitive processing of the default mode state.

A third factor to consider when testing the external validity of laboratory QEEG findings is the particular QEEG metric utilized to index brain functioning. The majority of previous studies using mobile EEG to study brain function have utilized event-related potential (ERP) measures of averaged stimulus-locked EEG activity [5,12,17,19,20,21,22,25,27,40], with few studies examining tonic or event-related spectral power [8,16,23,24,26]. Other studies have used measures of spectral power or functional connectivity to study the technical performance of mobile EEG equipment rather than brain function per se [7,9]. Here, we utilized EEG spectral power as the QEEG metric because previous research has demonstrated correlations between EEG oscillations within three frequency ranges (theta: 4–7 Hz, alpha: 8–13 Hz, low beta: 14–20 Hz) and the neurocognitive states engaged by the PASAT and resting state task [35,36,37,38,41,42,43,44,45,46,47,48,49,50].

Finally, we expected that a moderate influence of background environment would likely be small and idiosyncratic. Thus, to ensure sufficient sensitivity to both real and null effects, we employed null hypothesis significance testing (NHST) augmented by Bayesian model selection via use of Bayes factors. We observed EEG power effects typical of the neurocognitive states engaged by the PASAT and resting state task, but did not observe major EEG power differences between the two background recording environments for all three frequency bands. These results support the external validity of the EEG spectral power metric for complex and default neurocognitive states. (Note: This manuscript is based in part on an unpublished Master’s thesis submitted by the first author to Texas State University [51].)

## 2. Materials and Methods

### 2.1. Participants

All procedures were approved by the Institutional Review Board at Texas State University. For this study, 26 young adults were recruited through Texas State University’s SONA System recruiting pool. However, resting task EEG data from 5 participants were lost due to equipment malfunction; thus data from *n* = 21 participants were retained for the resting task. EEG data for the PASAT was also lost for 3 of the 5 participants, with lost resting task data due to equipment malfunction; thus data from *n* = 23 participants were initially available for the PASAT task. However, analysis of PASAT EEG requires a control condition for comparison, which here was the resting state eyes open condition (see Section 2.8 Statistical Analysis, below). Thus, the data for the 2 participants with complete PASAT data but missing resting state condition data were also excluded from the analysis to yield a final sample of *n* = 21 participants (mean age = 19.52, 95% CI = (18.60–20.45), 4 males and 17 females) for both the PASAT and the resting task. The participants were given course credit for participation in the study. All participants gave written informed consent in accordance with the Declaration of Helsinki. The Texas State University Institutional Review Board approved this study.

### 2.2. Background Recording Environments

Participant task performance and EEG signals were recorded in two environments within a single experimental session on the same day: our lab on the Texas State University campus and an outdoor area just outside the lab building (Figure 1), with the order of recordings within the two environments balanced across participants (for additional information about balancing, see Section 2.6 General Procedure, below).

The laboratory was a brightly lit, quiet, and stable environment maintained at a comfortable room temperature (approximately 21–22 °C per Texas State University policy). Participants sat in an office chair during task performance and EEG recording (Figure 1a), with a field of view consisting of stationary objects (e.g., office furniture, computer equipment, table, etc.). Only the researchers were present in the lab during the experimental session.

In the outdoor environment, participants sat on a metal bench during task performance and EEG recording (Figure 1b). In contrast to the laboratory, the outdoor environment was more complex and dynamic than the laboratory. The outdoor area contained trees and plants, buildings, small animals (birds, squirrels), people, other objects (e.g., bicycles, maintenance vehicles), etc., with all these elements at times stationary or in motion. Thus, the perceptual fields of the participants were filled with a multitude of potential visual and auditory distractions from participant task performance. However, outdoor sessions mostly took place in between class breaks where there was a minimal presence of people traveling through the outdoor area from one class to another.

In addition, the outdoor and laboratory environments also differed in terms of ambient lighting and other weather-related variables (Table 1). Outdoor experimental sessions always occurred during daylight hours, with most of the sessions occurring mid-day to early afternoon; this yielded a brightly sunlit environment on a clear day. (All sessions occurred over a period of time from October 2019 through February 2020). However, the ambient lighting was affected by cloud cover, which varied from session to session while remaining relatively stable within a session. On average, there was 43% cloud cover (i.e., partly cloudy) during any given session, with a corresponding decrease in ambient lighting relative to a clear day. We assume that atmospheric pressure and relative humidity were similar to the lab, given that these variables are largely driven both inside and outside campus buildings by the local weather. Nonetheless, on average, outdoor temperature was slightly lower than inside the laboratory (Table 1), although the latter temperature was well within the 95% CI of the average outdoor temperature. Finally, unlike the laboratory, the air of the outdoor environment typically exhibited a gentle breeze (4 m/s on average; see Table 1).

### 2.3. PASAT

In the PASAT task, participants listened to a human voice speak a single digit once every 3 s and then mentally added the most recent pair of digits together. For example, if the first two digits were “4” and “8”, then the participant should have written “12”. If the next digit was “2”, then the participant should have written “10”, as that is the new sum of 8 and 2. The auditory stimuli were presented to participants at a comfortable listening level via headphones from prerecorded audio files. Open back headphones (Model HD-6XX, Sennheiser, Wedemark, Germany) were used to ensure that participants would still be able to hear the background sounds within the natural environment. The open back headphones offered no noise cancellation while still providing adequate sound of the task stimuli to the participants.

Participants were given one of two prespecified versions of the task in each background recording environment, with the two versions of the task balanced between environments across participants (for additional information about balancing, see Section 2.6 General Procedure, below). Each PASAT consisted of 60 trials, with participants writing their answers on a premade score sheet. Each participant’s PASAT score was then computed as the total number of correct trials (max score = 60); this was computed separately for each performance environment. The participants were given a brief practice run in the first environment before they began the task and asked again in the second environment if they needed to reconfirm the task procedure. No feedback about task accuracy was given to participants during PASAT performance. PASAT performance lasted approximately 3 min. (We note that EEG recording for the PASAT lasted this 3 min plus approximately another 7 min due to technical recording limitations; see Section 2.5 EEG Recording, below).

### 2.4. Resting State Task

Participants underwent EEG recording while seated with eyes either closed or open over two separate blocks. Each resting state block lasted 10 min (with the first 5 min of EEG recordings retained for data analysis due to technical recording limitations; see Section 2.5 EEG Recording, below). In both environments, participants were verbally instructed to fixate their eyes on an arbitrary visual location in front of them for the duration of the task. Participants were also instructed to remain relaxed, alert, and awake, and to minimize eye movements/blinks during the recording in both environments. For each participant, the order of eyes open/closed periods was the same in each environment, but this order was balanced between environments across participants (for additional information about balancing, see Section 2.6 General Procedure, below).

### 2.5. EEG Recording

Continuous electroencephalographic brain activity was recorded using an OpenBCI Cyton mobile EEG amplifier (OpenBCI, New York, NY, USA) powered by a small 3.7 V 500 mAh, lithium ion polymer battery (Adafruit, New York, NY, USA); see Figure 2a,b. EEG recording parameters followed standard recommendations [52]. EEG signals were measured from sintered Ag/AgCl electrodes placed on the scalp at 16 international 10–20-electrode locations (Figure 2c). One additional electrode was affixed below the left eye to measure electrooculographic (EOG) activity (eye movements and blinks). The recording electrodes were embedded in a stretch lycra cap placed on the participants’ heads, with the electrode cap assembled in our lab from a commercially available electrode cap assembly kit (EasyCap, Herrsching, Germany).

The EEG recording cap was connected to the amplifier unit via a custom-made connector constructed in our lab (see Figure 2a,b). The amplifier was wirelessly connected to a USB dongle plugged into a recording laptop. The amp/USB connection used a high-power Bluetooth signal to control the device via the OpenBCI graphics user interface (GUI) recording and control software. EEG data were recorded to a Secure Digital (SD) card inside the EEG amplification unit. Raw data files were digitally stored in hexadecimal format on the SD card, and then later converted to decimal format text files via the OpenBCI GUI. The OpenBCI amplifier initially samples EEG signals at 125 Hz before automatically resampling them to 250 Hz; data were further resampled to 256 Hz during hexadecimal-to-decimal conversion of the data files. All electrode impedances were kept below 5 kΩ during recording. EEG signals were recorded with respect to a vertex (CZ) reference and a ground placed at AFZ.

We planned to record and analyze EEG data over the first 5 min of task performance for the resting task and first 3 min for the PASAT in order to eliminate any effects of mental fatigue due to extended time-on-task. The OpenBCI GUI records data for preset periods of time in 5 min steps. However, we experienced a recording limitation in which the Cyton unit recorded for a time length 1–3 min shorter than specified in the GUI. We overcame this limitation by recording EEG for approximately 10 min for each session and task. We then only retained the necessary number of minutes of EEG data for each task analysis.

### 2.6. General Procedure

Upon arrival, participants were informed of the procedure for the experiment before consent and equipment setup. Depending on the order of experimental session balancing, participants began the experiment either in the lab or in the outdoor environment (Figure 1). Participants then performed the resting task and the PASAT, with the order of performance of the two tasks within each environment balanced across participants. For each participant, the order of task performance was the same in each environment, but this order was balanced between environments across participants. In this study, the balancing of all relevant variables (environment, resting state condition, PASAT version, and task order) was achieved using a predetermined order list. Participants were then assigned to the next available order on the list when they arrived at the experimental session. Due to the need to exclude some participant data caused by equipment malfunction (see Section 2.1 Participants), all balancing in this study was approximately achieved in the final retained data set (although deviations from the ideally balanced orders were small).

The equipment and participant were then moved to the other recording location and the process repeated. We note that how quickly the EEG recording started in each environment depended on the order of the recordings. The recording in the first environment of the session occurred after the EEG setup, which lasted approximately 30–40 min. The recording in the second environment of the session only lasted approximately 10–15 min, which also involved a basic setup of the recording computer, task instruction reminders, and a check of EEG setup/connection status. However, the balancing of the order of the environments across participants should have controlled for any differential adaptation effects due to this time difference.

### 2.7. EEG Preprocessing and Spectral Power Analysis

Converted EEG data files were imported and processed within MATLAB computing software (MathWorks, Natick, MA, USA). Resting state and PASAT continuous EEG data were each decomposed into 1-second baseline-corrected epochs with 50% overlap, yielding 600 epochs and 180 epochs for the 5 min (300 s) and 3 min (180 s) of retained data for each task, respectively (see Section 2.5 EEG Recording, above). EEG data were then transformed to a digitally linked mastoids reference. Artifacts were removed from the EEG data using standard techniques [52], including automatic removal of epochs contaminated by eye blinks/movements (according to a ±50 mV rejection threshold applied to the left EOG channel) and manual removal of epochs contaminated by electromyographic (EMG) muscle activity, head motion, and other signal artifacts. Due to a high prevalence of motion artifacts resulting from the increased mobility of participants given the use of a mobile EEG system, artifact rejection was focused on maximizing the number of available epochs at relevant electrodes of interest (sites FZ and OZ; see Figure 2 and Section 2.8 Statistical Analysis, below) rather than across all electrodes across the scalp. For the resting task, the average number of epochs remaining after artifact removal was *M* = 467, 95% CI = (440–493) for the laboratory environment and *M* = 447, 95% CI = (417–479) for the outdoor environment. For the PASAT, the average number of epochs remaining after artifact removal was *M* = 229, 95% CI = (193–264) for the laboratory environment and *M* = 237, 95% CI = (206–268) for the outdoor environment. There were no statistical between-environment differences in retained trial numbers for either task (*ps* < 0.332, *B*_01_ > 2.76; see Section 2.7 Statistical Analysis for description of test statistics). Faulty EEG channels were replaced using an EEGLAB-based spherical spline interpolation algorithm [53] applied to the remaining channels (mean number of FZ channel interpolations = 0.14, 95% CI = (0–0.30); mean number of OZ channel interpolations = 0.19, 95% CI = (0.02–0.36)).

We computed EEG spectral power from the artifact-free EEG data via Fast Fourier Transform with a 1-second Hamming window. EEG power values were converted to decibel (dB) units. Following previous EEG studies of resting mental states, attention, working memory, and executive function [35,37,38,45,46,47,49,50], we quantified EEG power as the average power at frontal site FZ and at posterior site OZ within three separate frequency bands (theta: 4–7 Hz; alpha: 8–13 Hz; low beta; 14–20 Hz). EEG power was quantified for each participant and environmental condition, and for the resting task, each resting state condition (eyes closed, eyes open).

### 2.8. Statistical Analysis

We statistically analyzed behavioral and EEG power data using null hypothesis significance testing (NHST) augmented by Bayesian model selection via use of Bayes factors. NHST was used to compare the probability distribution of data described by an alternative hypothesis *H*_1_ of the presence of an experimental effect to the probability distribution described by the null hypothesis *H*_0_ of no effect. NHST was achieved using ANOVAs and Pearson correlation coefficients, where the associated F-values were used as an input to Bayes factors in order to compute direct probabilistic measures of the evidence for each hypothesis [54]. This is useful to adjudicate non-significant NHST outcomes, which cannot be readily interpreted as evidence for the null hypothesis [55].

#### 2.8.1. Null Hypothesis Significance Testing (NHST)

For the resting task, we applied 2 × 2 × 2 repeated measures analysis of variance (ANOVA) with factors of Environment (Laboratory, Outdoor), Resting State (Eyes Closed, Eyes Open), and Electrode (FZ, OZ). Following standard convention, we used a statistical significance criterion of *p* < 0.05. ANOVAs were conducted separately for each EEG power frequency band.

In order to assess EEG power reflecting cognitive engagement during the PASAT, a comparison condition is needed in which cognition is relatively disengaged. We utilized the eyes open resting state as this comparison condition. Hence for the PASAT task, we applied 2 × 2 × 2 repeated measures ANOVAs with factors of Environment, Task (PASAT, Eyes Open Resting State), and Electrode with a statistical significance criterion of *p* < 0.05. ANOVAs were conducted separately for each EEG power frequency band. Additionally, we analyzed PASAT performance scores via one-way repeated measures ANOVA with a within-participant factor of Environment.

Finally, the across-participant relationships between EEG power and the six outdoor environment weather-related variables listed in Table 1 (Temperature, Relative Humidity, Atmospheric Pressure, Wind Speed, Time of Day; see Section 2.2 Background Recording Environment) were assessed via Pearson correlation coefficients with a statistical significance criterion of *p* < 0.05 after correction for multiple comparisons across the six separate tests.

#### 2.8.2. Bayes Factor Analysis

A Bayes factor is the marginal likelihood ratio of two hypotheses (null hypothesis, alternative hypothesis), each specified as a statistical model, which quantifies the degree to which the data have increased or decreased the odds of one hypothesis relative to the other [54]. This then allows the determination of the probability of (“evidence for”) each hypothesis given the data. Here, we converted repeated-measures ANOVA information (*F*-values, degrees of freedom, and sample size) and Pearson correlation coefficients (via their associated *F*-values and degrees of freedom) into Bayes factors *B*_01_ in favor of the null hypothesis using an established formula based on the Bayesian Information Criterion (BIC) approximation of the Bayes factor [56,57,58,59]. This BIC-based method has the advantage that it does not require the specification of prior distributions [59]. Bayes factors in favor of the alternative hypothesis were then calculated as *B*_10_ = 1/*B*_01_. We interpreted the strength of support for a hypothesis as indicated by a Bayes factor according to the Jeffreys’ scale [60]: weak, 1 ≤ *B* < 3.16; substantial, 3.16 ≤ *B* < 10; strong, 10 ≤ *B* < 31.62; very strong, 31.62 ≤ *B* < 100; decisive, *B* ≥ 100. Following [56,57], we computed the probability of the null hypothesis given the data as *P*(*H*_0_|Data) = *B*_01_/(1 + *B*_01_) and the probability of the alternative hypothesis given the data as *P*(*H*_1_|Data) = 1 − *P*(*H*_0_|Data).

#### 2.8.3. Effect Size Analysis

As an extra step to assess the external validity of the present data, we conducted an analysis in which the sizes of the main effects for each task were compared to the effect sizes obtained from a simple meta-analysis of several laboratory EEG studies that utilized these tasks. For simplicity, we focused our analysis on each task’s main effect that statistically quantified the activation of the specific neurocognitive states engaged by each task. For the resting task, this was the main effect of Resting State; for the PASAT, this was the main effect of Task. It was not our intent to perform an exhaustive meta-analysis of the previous literature, but only to identify enough studies to obtain a reasonable estimate of the general effect sizes observed in the laboratory. Moreover, we only utilized studies with design and analysis structure that could be easily compared to that of the present study. We chose studies to include in the meta-analysis according to the following criteria: (1) a study utilized a resting state task, an arithmetic task, and/or the PASAT, (2) a study used a repeated-measures contrast to compare EEG power across experimental variables similar to the present study (resting task: eyes open versus eyes closed; PASAT: arithmetic performance versus rest or similar control condition); and (3) a study explicitly reported the information necessary to compute effect size (means, standard deviations/errors, and/or values of inferential statistics). Using these criteria, we identified 7 studies using the resting task, with 4 of these studies measuring theta and beta power [37,47,61,62], and all 7 studies measuring alpha power [35,36,37,38,47,61,62]. We also identified 9 studies using an arithmetic task or the PASAT, with 5 of these studies measuring theta power [63,64,65,66,67], 7 measuring alpha power [63,64,65,67,68,69,70], and 3 studies measuring beta power [66,69,71]. For ease of comparison across studies, the Hedges’ g effect size statistic was used because it reflects a standardized difference that is corrected for bias [72,73]. This statistic was calculated for all studies using appropriate formulas to compute mean effect sizes and 95% confidence intervals (CIs) across studies [72,73,74]. We then compared this to Hedges’ g computed for the present study’s corresponding main effects of interest, with this comparison performed separately for the laboratory and outdoor EEG power, and also after collapsing across these two conditions.

## 3. Results

All empirical data and MATLAB data analysis scripts are available online at the Texas State University Data Repository (https://dataverse.tdl.org/dataverse/brainscieeg; accessed on 20 February 2021).

### 3.1. Resting Task EEG Power

Figure 3 shows spectrum plots of resting state EEG power; mean resting state EEG power values for each frequency band of interest are listed in Table 2. The resting state spectrum demonstrated the typical greater eyes closed versus eyes open power over all three frequency bands, with the largest difference present within the alpha band. These observations were supported by the statistical analysis (Table 3), which showed a significant main effect of Resting State for all three frequency bands, with strong to decisive Bayesian evidence for these differences in the theta- and alpha-bands and weaker evidence for the beta-band difference. Main effects of Electrode were also significant for these bands with strong to decisive Bayesian evidence (Table 3), indicating greater EEG power over the posterior versus frontal scalp for the alpha and beta frequency bands and vice versa for the theta band (Table 2).

Importantly, there was a significant main effect of Environment for beta-band power (Table 3), with slightly greater power for the outdoor versus laboratory environment (Table 2). However, the Bayesian evidence for this effect was weak. There were no significant main effects of Environment for the theta and alpha bands (Table 3), with weak Bayesian evidence for the theta band null effect and strong evidence for the alpha band null effect.

Finally, we performed a correlational analysis of the across-participant relationships between the six outdoor environment weather-related variables listed in Table 1 and the outdoor versus laboratory resting state EEG power difference after collapsing across the resting state and electrode factors. We found no significant correlations for any frequency band after correcting for multiple comparisons across the six separate tests, all *ps* = 1.00, *B*_01_ = 4.47, *P*(*H*_0_|Data) = 0.82.

### 3.2. PASAT EEG Power and Behavioral Performance

Figure 4 shows spectrum plots of PASAT EEG power; mean PASAT EEG power values for each frequency band of interest are listed in Table 4. Statistical analysis (Table 5) comparing EEG power between the PASAT and the resting state eyes open comparison condition demonstrated a statistically significant main effect of Task for the alpha band. PASAT alpha-range power was significantly lower than then resting state eyes open comparison condition, with strong Bayesian evidence for this effect. This difference is in the direction expected to occur during the complex cognitive states (attention, working memory, executive control) engaged by the PASAT (see Section 4 Discussion, below).

Main effects of Electrode were significant for all three frequency bands with strong to decisive Bayesian evidence (Table 5). These effects indicated greater EEG power over the frontal versus posterior scalp within the theta band and greater EEG power over the posterior versus frontal scalp within the alpha and beta bands (Table 4). However, the beta-band Electrode main effect was accompanied by a significant Task × Electrode interaction. Decomposition of this interaction revealed that the posterior versus frontal beta-band power difference (2.23 µV) was large for the PASAT, *F*(1,20) = 24.32, *p* < 0.001, *B*_01_ = 927.41, *P*(*H*_1_|Data) = 1.00, but smaller (0.73 µV) and statistically weaker for the eyes open resting state, *F*(1,20) = 4.96, *p* < 0.038, *B*_10_ = 2.23, *P*(*H*_1_|Data) = 0.69.

Notably, although the PASAT spectrum displayed small qualitative differences between background recording environments (Figure 4), there was no statistical support for these differences when comparing EEG power between the PASAT and the eyes open resting state (Table 5). There were no significant main or interaction effects involving the factor of Environment for all three frequency bands (Table 5), with most of these null findings having strong Bayesian evidence in their favor and few with weak evidence in their favor. One null effect (a theta-band Environment × Task interaction) approached significance (*p* < 0.084) with weak Bayesian evidence; further decomposition of this interaction yielded no significant outcomes (*ps* < 0.101, *B*_01_ = 1.08 to 4.23).

We also performed a correlational analysis of the across-participant relationships between the six outdoor environment weather-related variables listed in Table 1 and the difference between the outdoor and laboratory between-task differences (i.e., PASAT minus eyes open resting state) in EEG power. We found no significant correlations for any frequency band after correcting for multiple comparisons across the six separate tests, *ps* = 0.12 to 1.00, *B*_01_ = 1.22 to 4.47, *P*(*H*_0_|Data) = 0.55 to 0.82.

Finally, no PASAT behavioral performance differences were observed between the laboratory (PASAT score = 44.14, 95%CI (35.62–52.67)) and outdoor (PASAT score = 46.86, 95%CI (39.50–54.22))) recording environments, *F*(1,20) = 1.22, *p* < 0.28, *B*_01_ = 2.47, *P*(*H*_0_|Data) = 0.71.

### 3.3. Resting Task and PASAT EEG Power Variability Analysis

Given the fact that the outdoor environment was more complex and less stable than the laboratory (see Materials and Methods Section 2.2 Background Recording Environments), we investigated if the variability of EEG power was also affected by the recording environment. We computed the standard deviation of EEG power for each task condition, environment, electrode and participant. We used standard deviation as the variability metric because it has the same initial units as spectral power; these values were then converted to decibels. For simplicity of interpretation, we collapsed EEG power variability values across electrodes for the statistical analysis.

The pattern of results from these analyses paralleled the EEG power findings; see Table 6 and Table 7. Resting state beta-range power variability was significantly larger for the outdoor versus laboratory environment with weak Bayesian evidence. Resting state theta-, alpha-, and beta-range power variability was larger for the eyes closed versus eyes open conditions with strong to decisive Bayesian evidence. Alpha-range power variability was significantly smaller during the PASAT versus the eyes open resting state with weak Bayesian evidence. A theta-band main effect of Task and an Environment × Task interaction both approached significance (*ps* < 0.082) for the PASAT, but the corresponding Bayesian evidence for these outcomes was weak. No other environmental effects of EEG power variability were significant for either task (with weak to strong Bayesian evidence in favor of the null hypothesis).

Finally, we performed a correlational analysis of the across-participant relationships between the six outdoor environment weather-related variables listed in Table 1 and (1) the difference between outdoor and laboratory resting state EEG power variability after collapsing across the resting state and electrode factors, and (2) the difference between outdoor and laboratory between-task differences (i.e., PASAT minus eyes open resting state) in EEG power variability. We found no significant correlations for any frequency band after correcting for multiple comparisons across the six separate tests, *ps* = 0.53 to 1.00, *B*_01_ = 3.61 to 4.47, *P*(*H*_0_|Data) = 0.78 to 0.82.

### 3.4. Effect Size Analysis

Table 8 shows the estimated observed sizes of the Resting State main effect (Eyes Closed versus Eyes Open) for the resting task. The table lists effect sizes for the laboratory and outdoor conditions separately, and after collapsing across these two conditions. Given the absent to weak evidence for statistical interactions between the ANOVA factors of Resting State and Electrode (see Table 3), observed effect sizes were derived from EEG power collapsed across sites FZ and OZ. Table 8 also shows the effect sizes estimated from the resting state studies we included in the meta-analysis (see Section 2.8.3 Effect Size Analysis, above). For these studies, the effect sizes were based on reported statistics derived from EEG power at either posterior scalp sites or collapsed across the whole scalp. In general, the observed effect sizes were in the same direction as and smaller than the meta-analytic effect sizes. This observation is supported by the fact that the confidence intervals of the observed and meta-analytic effects did not overlap.

Table 9 shows the estimated observed effects sizes for the Main Effect of Task (PASAT versus Resting Task Eyes Open) for the PASAT. The table lists the effect sizes for the laboratory and outdoor conditions separately, as well as after collapsing across these two conditions. Given the absent evidence for statistical interactions between the ANOVA factors of Task and Electrode (see Table 5), observed effect sizes were derived from EEG power collapsed across sites FZ and OZ for theta- and alpha-band power. In the case of beta-band power, however, separate assessments were performed for the two scalp sites due to previous observations of distinct frontal beta increases and posterior beta decreases during arithmetic performance [66,69,71] (for further discussion, see Section 4.2 Discussion: PASAT below). Table 9 also shows effect size estimates derived from the mental arithmetic/PASAT studies we included in the meta-analysis (see Section 2.8.3 Effect Size Analysis, above). For these studies, the effect sizes were based on reported statistics derived from EEG power at frontal scalp sites for theta-band power, posterior scalp sites or collapsed across the whole scalp for alpha-band power, and at frontal and posterior sites separately for beta-band power.

Table 9 shows that the overall pattern of effect sizes for the PASAT was more complex than for the resting task. The presently observed theta-band effect sizes were small in magnitude and mostly negatively valued, whereas the corresponding theta-band meta-analytic effect was positive and large in magnitude. The presently observed alpha-band effect sizes were negative, consistent with the direction of the corresponding alpha-band meta-analytic effect, but with larger magnitudes than the latter. These theta- and alpha-band observations are supported by the fact that the confidence intervals of the observed and meta-analytic effects did not overlap. Furthermore, frontal beta power effect sizes were negative for the outdoor and collapsed conditions, whereas the corresponding frontal-beta band meta-analytic effect size estimate was positive, with non-overlapping confidence intervals between the observed and meta-analytic effects. In contrast, the frontal beta power effect for the laboratory condition, though positive, was near zero. Moreover, posterior beta power effect sizes for the laboratory, outdoor, and collapsed conditions were positive, whereas the corresponding posterior beta-band meta-analytic effect size was negative, with non-overlapping confidence intervals for the two sets of effects.

## 4. Discussion

In the present study, we explored the external validity limits of laboratory-based QEEG findings by examining the influence of physical environment background on complex and simple neurocognitive states as, respectively, engaged by the PASAT and resting state task. Each participant performed these tasks in a controlled laboratory environment (closed space, minimal noise, static temperature and atmosphere) and a mildly uncontrolled outdoor environment (open space, background noise, weather changes). Brain activity associated with these neurocognitive states was quantitatively indexed via computation of EEG spectral power. We observed a weak between-environment beta band power effect for the resting task, with greater beta power in the outdoor versus laboratory environment, but no other reliable spectral power differences between the two background recording environments in all spectral power bands for either task.

### 4.1. Resting Task

The resting state task engages a default mental state, characterized by internal cognitive processing (conceptualization, episodic working memory, unconstrained verbally mediated thoughts), endogenously directed attention, and the monitoring of the body, external environment, and emotional states [39,75,76,77,78]. These mental states are correlated with the oscillatory activity of several resting state brain networks distributed across multiple brain regions [76], where such activity is disengaged during active goal-directed behavior [79]. Importantly, these resting state networks are associated with unique patterns of wideband (0.1–100 Hz) EEG power [76], although network activity in the theta, alpha, and beta band ranges predominates [80]. Moreover, the resting EEG power spectrum is modulated according to whether an individual maintains the resting state with eyes open and closed. Most studies consistently find greater theta, alpha, and beta spectral power during eyes closed versus eyes open states [35,36,37,38,41,47,61,62], differences that in part reflect a transition from “cortical idling” in the absence of visual or cognitive stimulation to active perceptual and cognitive engagement [80]. We also observed this eyes closed versus eyes open activity pattern across both environments within all three frequency bands in the present study. The observation of greater eyes closed versus eyes open EEG power for both environments supports the external validity of laboratory observations of the resting state EEG power across all three frequency ranges.

One concern with this observation is the degree to which it reflects the typical transition from one default mode state to another, or if it reflects environmental factors as well. For example, eyes open alpha rhythms can also be can be blocked by changes in visual stimulation via eye movements as participants fixate attention on different visual elements in their environment [81]. Given that the outdoor environment provided a more complex and dynamic scene than the laboratory (see Section 2.2 Background Recording Environments), some proportion of the alpha blocking observed in the eyes open resting state may have been due to eye movements rather than a general transition away from an idle cortical state. In addition, resting state EEG power responses in the outdoor environment may also reflect higher arousal levels due to embarrassment at being seen by others during the experiment while wearing an awkward-looking EEG cap. Nevertheless, we did not find statistically meaningful between-environment differences for resting state theta and alpha EEG power or EEG power variability, nor did we find such differences to correlate with the six weather-related variables we indexed in the outdoor environment (see Section 2.2 Background Recording Environments). This suggests that such visual factor or general arousal differences between the two environments did not contribute much variance to our present EEG power measurements in the theta and alpha frequency bands.

However, the present observation of greater resting state beta-band EEG power for the outdoor versus indoor environments suggests that certain resting state processes can be affected by the environment. In general, the frequency of EEG oscillation reflects the size of the recurrent neural network mediating the oscillation [82,83], with higher frequency oscillations reflecting small (local) network activity and low frequency oscillations reflecting large (global) network activity. Thus, the present between-environment beta-band difference may reflect differences in local network processing. Moreover, given that this effect did not differ across the scalp (as indicated by a null Environment × Electrode effect; see Section 3.1. Resting Task EEG Power, above), it is possible that these beta-band differences are present in local networks spread across the cortex rather than in specific brain regions. Furthermore, given that the present beta-band resting state EEG power effects were observed in the context of greater beta-band EEG power variability in the outdoor environment (see Section 3.3 Resting Task and PASAT EEG Power Variability Analysis), it is possible that the beta-band EEG power effect reflects more variable activity of these local networks in response to the unstable outdoor environment. However, testing these possibilities requires additional EEG source localization analysis to account for the known volume conduction and spatial dispersion of cortical EEG signals as they travel through the head from the cortex to the scalp [84,85]. Such analysis is beyond the scope of the present study and is a goal for future research.

We must note that the significant beta-band effect was weak according to Bayesian evidence criteria (see Section 3.1. Resting Task EEG Power, above). Given the weakness of the effect, we cannot rule out the alternative possibility that it merely reflects residual EMG contamination of the EEG signal that was not entirely removed by our artifact rejection procedure (see Section 2.7 EEG Preprocessing and Spectral Power Analysis, above). Muscle activity generates wide-band high frequency gamma (25–100 Hz) EEG signals that overlap with the beta range [82,86], and it is possible that participants were less relaxed and exhibited more muscular tension and movement in the outdoor environment relative to the laboratory environment. This interpretation would be consistent with our observation that between-environment differences in resting state beta power did not correlate with the six weather-related variables we indexed in the outdoor environment. Although, the likelihood of this alternative possibility is low given that we focused on the lower beta band (14–20 Hz) that is less susceptible to EMG contamination, future research is needed to replicate the present beta-band power finding.

We also performed an effect size analysis in which we compared the observed sizes of the resting task’s primary, neurocognitively relevant main effect (differences in eyes closed versus open resting states) to the effect sizes obtained from a simple meta-analysis of several laboratory EEG studies that utilized this task. We found the directions of the presently observed resting state primary effects were consistent with the directions of the meta-analysis effects. This was case for the EEG data recorded in the laboratory and outdoor environments, which supports the external validity of laboratory resting state EEG findings. However, we also found the presently observed effects sizes to be smaller than the meta-analytic effect sizes obtained from laboratory-recorded data. Given that smaller effect sizes were observed for the present resting state data recorded in both environments, it is likely that this reflects a performance difference between traditional laboratory and mobile EEG technology. Mobile EEG technology is known to exhibit lower signal-to-noise ratios than traditional EEG systems [9,13,15,16], and it is possible that the smaller effect sizes observed in the present study reflect this factor. This is a topic for further research.

### 4.2. PASAT

The PASAT is an arithmetic task that requires numerical information to be represented, retained, and transformed within the mind. The task engages a complex combination of cognitive functions, including number sense, attention, short- and long-term memory, and executive control [32,33,34]. Similar to the resting task, the mental states of the PASAT are correlated with the oscillatory activity of several brain networks distributed across multiple brain regions [76,79,87]. Unlike the resting task, however, these networks are engaged during active goal-directed behavior, with their activity anti-correlated with the activation of the resting state default mode network [79]. Relative to a rest or baseline state, EEG spectral power patterns elicited during simple arithmetic tasks typically exhibit increases in frontal theta and/or beta power [63,64,65,66,67,71], decreases in posterior beta power [66,69], and decreases in posterior or scalp-wide alpha power [63,64,65,68,69,70]. The frontal theta and beta power changes are thought to reflect executive engagement, active cognition and concentration, and/or functional binding [48,66,67,88,89]. The alpha power changes likely reflect a transition from “cortical idling” in the absence of visual or cognitive stimulation to active perceptual and cognitive engagement [64,65,69,90], similar to the alpha power decrease seen during the resting task when transitioning from eyes closed to eyes open states.

In the present study, we observed similar alpha power decreases during PASAT performance relative to the resting task eyes open control condition (see Section 3.2 PASAT EEG Power and Behavioral Performance, above). This finding was observed in the context of a null effect of background recording environment for the alpha power bands and an absence of a correlation between the six weather-related variables we indexed in the outdoor environment and the difference between the outdoor and laboratory between-task differences in EEG power. Furthermore, we compared the observed sizes of the PASAT’s primary, neurocognitively relevant main effect (differences in PASAT versus eyes open resting state control) to the sizes of similar effects obtained from a simple meta-analysis of several laboratory EEG studies that utilized this task. This analysis yielded observed alpha-band effect sizes that were negative and large in magnitude, consistent with the meta-analytic alpha-band effects. These results support the external validity of these alpha EEG power effects as observed in the laboratory.

However, we did not observe any meaningful between-task differences in theta-power. It is unclear if the lack of these between-task differences is due to effects of environment or something specific to the PASAT. If environmental influences obscured the presence of a true theta power effect, then this should be detectable in the form of a main ANOVA effect of environment and/or an interaction of environment with other experimental variables. Yet, we did not observe any significant theta-band effects of environment (both from the ANOVAs and the Pearson correlations between EEG power and the six weather-related variables we indexed in the outdoor environment), and the Bayesian evidence for these null effects were mostly well-above chance; *P*(*H*_0_|Data) ranged from 0.72 to 0.82, save for an Environment × Task ANOVA interaction, *P*(*H*_0_|Data) = 0.48 (see Section 3.2 PASAT, above). In addition, the present effect size analysis showed that the observed theta-band effects were either positive but near zero in value, or negative valued. This is in contrast to the large positive-valued meta-analytic theta-band effect, but it is consistent with the inferential tests for theta-band EEG power during the PASAT. Regarding the possibility that these discrepant null theta-band task effects are specific to the PASAT, a survey of past EEG research literature yields little insight. Thus, far, surprisingly few EEG studies have focused directly on PASAT performance-elicited electrophysiological responses. Instead, most past studies have focused on using this task in conjunction with EEG to understand longitudinal changes in PASAT performance due to practice or brain injury [91,92,93]. The few studies that have focused on PASAT-elicited EEG responses found no theta band differences when PASAT performance is compared to some control condition (e.g., another active task or default resting states) [70,94]. Thus, while the present null theta power effects are in line with previous findings, and thus support a conclusion of external validity for these effects, this conclusion is tentative until the present theta power null finding is replicated by future research.

We also did not observe clear between-task differences in beta-band power. We did find a significant Task × Electrode interaction for beta-band power, but further analysis showed that this interaction reflected a positive difference between posterior versus frontal power that was stronger for the PASAT than the eyes open resting state. However, there were no significant between-task differences at either scalp location (see Section 3.2. PASAT EEG Power and Behavioral Performance). The effect size analysis showed a different pattern of task-related beta-band EEG power compared to the meta-analytic effect estimate obtained from previous studies. Task-related posterior beta-band EEG power was positive-valued, in contrast to the negative value of the corresponding meta-analytic effect. The effect size analysis also suggested a different pattern of task-related beta-band EEG power across environments. Task-related frontal beta-band power was positive-valued (though near zero) within the laboratory environment, in agreement with the positive value of the corresponding meta-analytic effect, yet was negative-valued within the outdoor environment. Nevertheless, statistical tests (ANOVAs, Pearson correlations) involving the factor of environment were non-significant with above-chance Bayesian evidence for these null effects (*P*(*H*_0_|Data) ranged from 0.70 to 0.82). We suggest that this complex pattern of discrepant results for the PASAT beta-band power could be due to two factors: (1) neurocognitive differences in arithmetic versus PASAT performance (the latter of which involves both mathematical and working memory cognition) or (2) a true difference in arithmetical neurocognition across different environments that was too small to be fully detected by our current experimental design or was otherwise obscured by other factors (for example, unmeasured environmental characteristics or a lower-signal to-noise ratio for mobile EEG). Determining which (if any) of these factors is responsible for the present observations is a topic for future research.

### 4.3. Study Limitations

Interpretation of the present observations must be tempered by consideration of the limitations of the present study. One limitation was the small sample size and resulting low statistical power. However, this limitation is somewhat mitigated by the use of Bayes factors, which allow us to estimate the probability of null and alternative effects. This allowed us to interpret non-significant NHST outcomes as evidence for the null hypothesis [55].

A second limitation of this study is that it has lowered internal validity with respect to the main factor of interest, the effect of background influences in the physical environment. This was in part by design because the objective of our study was to evaluate the external validity of QEEG research in different physical environments. We had no direct control of these influences in the outdoor environment, influences that fluctuated greatly. We had no control over levels of student activity on campus, noise levels, and weather activity. We did implement limited control over the time of day experimental sessions were conducted; the majority of participants were run during mid-afternoon hours when the campus was highly active. However, due to time constraints, not all experimental sessions were conducted at the same hours of the day and thus some sessions were conducted during periods of quiet campus activity. Thus, it remains possible that the present null effects of environment resulted in part from limited distractions in the outdoor environment that were not strong enough to influence task performance.

A third limitation of the present study is that measurement of the various characteristics of the two physical environments was limited. While we did not observe a direct relationship between several key weather variables (see Section 2.2 Background Recording Environments) and EEG power, there are multiple other characteristics of these environments that we did not assess. Thus, it is unclear to which the present results can be generalized to other environments.

A fourth limitation of the present study is that we only used two basic behavioral tasks that did not involve large-scale active movement to appraise neurocognitive functioning. Thus, we cannot generalize our findings to other tasks engaging similar cognitive processes as studied here; this is a topic for future research.

Finally, a fifth limitation of this study is that our population sample was entirely composed of young, college students. It is likely that their demographic characteristics (e.g., level of education, mathematical experience) influenced PASAT performance and electrophysiological outcomes. Investigating the external validity of the present QEEG metrics for other populations performing these tasks is a topic for future research.

## 5. Conclusions

In conclusion, the present study probed the validity limits of laboratory QEEG by using a mobile EEG system to record EEG signals from human participants while they performed two neurocognitive tasks (PASAT, resting state task) within a controlled laboratory environment and a moderately uncontrolled outdoor environment. Null hypothesis significance testing (NHST) showed significant EEG spectral power effects typical of the neurocognitive states engaged by these tasks (number sense, attention, memory, executive function), but only a beta-band EEG power difference between the two recording environments for the resting task. Bayesian analysis showed that the remaining null effects of environment were unlikely to reflect measurement insensitivities. The overall pattern of these results supports the external validity of laboratory EEG spectral power findings for the complex and default neurocognitive states engaged in moderately uncontrolled environments. They also serve to bolster the credibility of efforts to use mobile EEG systems to index neurocognitive performance in non-laboratory environments.

Human beings operate in a multitude of environments each day, which range drastically regarding the type and concentration of stimuli that are present, yet differences in environmental influences on cognition have been understudied. The present study used QEEG methods to only evaluate the effects of two particular physical environments using two specific cognitive tasks. Future mobile EEG research should study additional physical environments and tasks in order to further examine the external validity of QEEG and to better understand human neurocognition in real-world environments.

## Figures and Tables

**Figure 1 brainsci-11-00330-f001:**
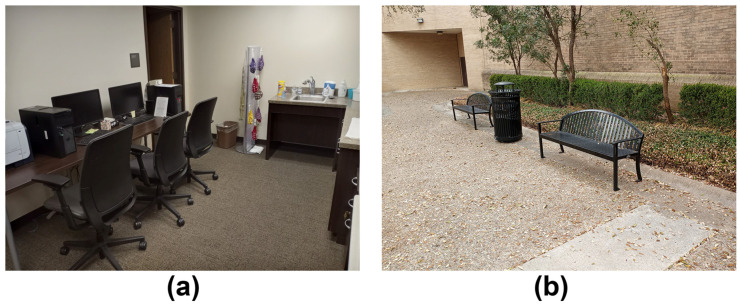
EEG recording environments: (**a**) Laboratory environment; (**b**) Outdoor environment.

**Figure 2 brainsci-11-00330-f002:**
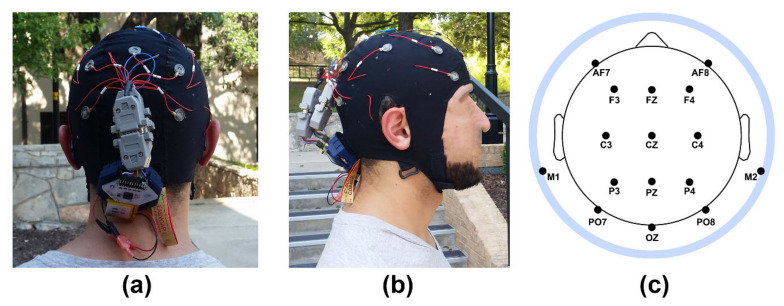
Mobile EEG recording setup: (**a**) Back view of EEG cap and recording amplifier. The amplifier was attached to the back of the cap via a Velcro patch and connected to cap leads via a custom-made connector; (**b**) Side view of EEG cap and recording amplifier. (**c**) Extended 10–20 scalp locations of EEG recording electrodes. Selected electrodes of interest (EOIs) for data analysis were sites FZ and OZ. Note that M1/M2 sites outside the radius of the head represent locations that are below the equatorial plane (FPZ-T7-T8-OZ plane) of the (assumed spherical) head model. Site LVEOG was located below the left eye approximately at the same latitude as sites M1 and M2.

**Figure 3 brainsci-11-00330-f003:**
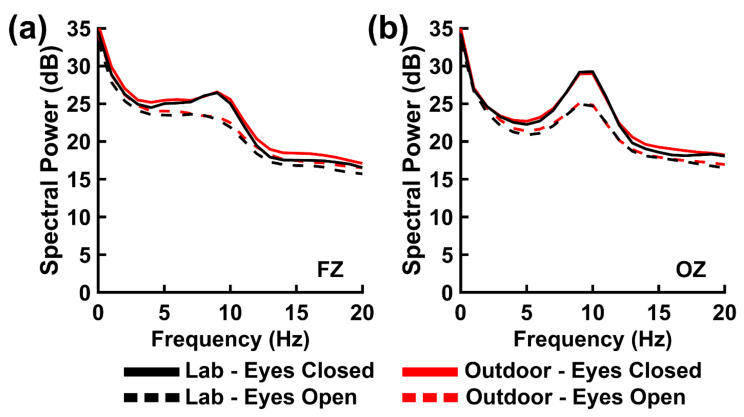
EEG spectral power (in decibels) at (**a**) frontal site FZ and (**b**) posterior site OZ for eyes closed (solid lines) and eyes open (dashed lines) resting states in the laboratory (black lines) and outdoor (red lines) environments.

**Figure 4 brainsci-11-00330-f004:**
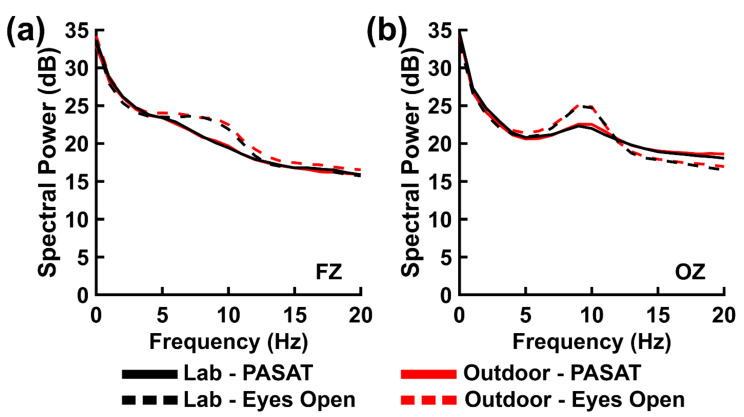
EEG spectral power (in decibels) at (**a**) frontal site FZ and (**b**) posterior site OZ for PASAT (solid lines), and eyes open resting states (dashed lines) for comparison, in the laboratory (black lines) and outdoor (red lines) environments.

**Table 1 brainsci-11-00330-t001:** Mean outdoor environment variables across experimental sessions.

Temperature	Relative Humidity	Atmospheric Pressure	Wind Speed	Cloudiness	Time of Day
19	57	742	4	43	14:00
(16–21)	(47–68)	(730–753)	(3–5)	(23–64)	(13:00–15:00)

Note: Temperature units: °C, Relative Humidity units: %, Atmospheric Pressure units: mm Hg, Wind Speed units: m/s, Cloudiness: % sky cover, Time of Day format: 24-hour clock. Parentheses show 95% CIs. Weather variable averages are based on information for the date and time of each experimental session obtained from an online weather data repository (Weather Underground, www.wunderground.com, accessed on 20 February 2021).

**Table 2 brainsci-11-00330-t002:** Mean resting state EEG power.

Frequency Band	Environment	Eyes Closed		Eyes Open	
		FZ	OZ	FZ	OZ
Theta	Laboratory	24.88	22.70	23.65	21.33
		(23.65–26.11)	(20.97–24.42)	(22.65–24.65)	(20.03–22.63)
	Outdoor	25.43	23.02	24.2	21.9
		(24.22–26.64)	(21.47–24.56)	(23.24–25.15)	(20.82–22.97)
Alpha	Laboratory	24.05	26.2	21.74	22.99
		(22.77–25.34)	(24.31–28.08)	(20.57–22.92)	(21.15–24.84)
	Outdoor	24.45	26.15	22.12	23.08
		(22.98–25.92)	(24.17–28.12)	(20.78–23.47)	(21.12–25.04)
Beta	Laboratory	17.46	18.61	16.65	17.62
		(16.40–18.52)	(17.29–19.93)	(15.65–17.65)	(16.42–18.81)
	Outdoor	18.27	19.18	17.33	17.82
		(16.97–19.56)	(18.02–20.34)	(16.24–18.42)	(16.60–19.04)

Note: All values are in decibels (dB); 95% CIs in parentheses.

**Table 3 brainsci-11-00330-t003:** Analysis of variance (ANOVA) for Resting Task EEG power.

Frequency Band	Effect	*F*(1,20)	*η* ^2^ _P_	*B* _01_	*B* _10_	*P*(*H*_0_|Data)	*P*(*H*_1_|Data)
Theta	ENV	1.79	0.08	1.86 ^†^	0.54	0.65	0.35
	RS	10.38 **	0.34	0.06	17.60 ^†††^	0.05	0.95
	ELEC	28.52 ***	0.59	0.00	2400.18 ^†††^	0.00	1.00
	ENV × RS	0.09	0.01	4.36 ^††^	0.23	0.81	0.19
	ENV × ELEC	0.02	0.00	4.53 ^††^	0.22	0.82	0.18
	RS × ELEC	0.00	0.00	4.58 ^††^	0.22	0.82	0.18
	ENV × RS × ELEC	0.60	0.03	3.37 ^††^	0.30	0.77	0.23
Alpha	ENV	0.25	0.01	4.03^††^	0.25	0.80	0.20
	RS	29.35 ***	0.60	0.00	2866.86 ^†††^	0.00	1.00
	ELEC	9.43 **	0.32	0.08	12.59 ^†††^	0.07	0.93
	ENV × RS	0.00	0.00	4.58 ^††^	0.22	0.82	0.18
	ENV × ELEC	0.39	0.02	3.74 ^††^	0.27	0.79	0.21
	RS × ELEC	3.84	0.16	0.72	1.38 ^†^	0.42	0.58
	ENV × RS × ELEC	0.06	0.00	4.45 ^††^	0.22	0.82	0.18
Beta	ENV	4.74 *	0.19	0.49	2.04 ^†^	0.33	0.67
	RS	5.32 *	0.21	0.38	2.60 ^†^	0.28	0.72
	ELEC	8.17 **	0.29	0.13	7.97 ^††^	0.11	0.89
	ENV × RS	0.38	0.02	3.77 ^††^	0.27	0.79	0.21
	ENV × ELEC	0.22	0.01	4.08 ^††^	0.25	0.80	0.20
	RS × ELEC	1.25	0.06	2.43 ^†^	0.41	0.71	0.29
	ENV × RS × ELEC	0.52	0.03	3.50 ^††^	0.29	0.78	0.22

ANOVA factor labels: ENV = Environment, RS = Resting State, ELEC = Electrode. *F*-value significance: * = *p* < 0.05, ** = *p* < 0.01, *** = *p* < 0.001. Bayes factor evidence strength: † = weak, †† = strong, ††† = very strong to decisive.

**Table 4 brainsci-11-00330-t004:** Mean PASAT EEG power.

Frequency Band	Environment	FZ	OZ
Theta	Laboratory	23.69	21.52
		(22.77–24.61)	(20.65–22.40)
	Outdoor	23.55	21.21
		(22.77–24.33)	(20.37–22.05)
Alpha	Laboratory	19.82	21.47
		(18.85–20.79)	(19.91–23.02)
	Outdoor	19.85	21.69
		(18.84–20.87)	(20.29–23.10)
Beta	Laboratory	16.77	18.84
		(15.70–17.85)	(17.36–20.33)
	Outdoor	16.6	19.00
		(15.62–17.59)	(17.74–20.27)

Note: All values are in decibels (dB); 95% CIs in parentheses.

**Table 5 brainsci-11-00330-t005:** Analysis of variance (ANOVA) for PASAT versus Resting Task Eyes Open EEG power.

Frequency Band	Effect	*F*(1,20)	*η* ^2^ _P_	*B* _01_	*B* _10_	*P*(*H*_0_|Data)	*P*(*H*_1_|Data)
Theta	ENV	0.61	0.03	3.34 ^††^	0.30	0.77	0.23
	TASK	1.11	0.05	2.60 ^†^	0.38	0.72	0.28
	ELEC	59.17 ***	0.75	0.00	410,005.21 ^†††^	0.00	1.00
	ENV × TASK	3.32	0.14	0.92	1.09 ^†^	0.48	0.52
	ENV × ELEC	0.05	0.00	4.48 ^††^	0.22	0.82	0.18
	TASK × ELEC	0.02	0.00	4.54 ^††^	0.22	0.82	0.18
	ENV × TASK × ELEC	0.05	0.00	4.45 ^††^	0.22	0.82	0.18
Alpha	ENV	0.18	0.01	4.17 ^††^	0.24	0.81	0.19
	TASK	14.07 ***	0.41	0.02	58.68 ^†††^	0.02	0.98
	ELEC	11.29 **	0.36	0.04	23.94 ^†††^	0.04	0.96
	ENV × TASK	0.03	0.00	4.50 ^††^	0.22	0.82	0.18
	ENV × ELEC	0.02	0.00	4.54 ^††^	0.22	0.82	0.18
	TASK × ELEC	1.68	0.08	1.96 ^†^	0.51	0.66	0.34
	ENV × TASK × ELEC	0.34	0.02	3.84 ^††^	0.26	0.79	0.21
Beta	ENV	0.66	0.03	3.25 ^††^	0.31	0.76	0.24
	TASK	1.02	0.05	2.72 ^†^	0.37	0.73	0.27
	ELEC	29.10 ***	0.59	0.00	2720.88	0.00	1.00
	ENV × TASK	1.31	0.06	2.36 ^†^	0.42	0.70	0.30
	ENV × ELEC	0.03	0.00	4.52 ^††^	0.22	0.82	0.18
	TASK × ELEC	7.01 *	0.26	0.20	5.12	0.16	0.84
	ENV × TASK × ELEC	0.89	0.04	2.90 ^†^	0.35	0.74	0.26

ANOVA factor labels: ENV = Environment, TASK = Task, ELEC = Electrode. *F*-value significance: * = *p* < 0.05, ** = *p* < 0.01, *** = *p* < 0.001. Bayes factor evidence strength: † = weak, †† = strong, ††† = very strong to decisive.

**Table 6 brainsci-11-00330-t006:** Average EEG power variability.

Frequency Band	Environment	Resting Task	Resting Task	PASAT
		Eyes Closed	Eyes Open	
Theta	Laboratory	24.51	23.09	23.03
		(23.05–25.97)	(21.90–24.28)	(22.17–23.89)
	Outdoor	25.03	23.94	22.78
		(23.74–26.32)	(23.02–24.86)	(22.08–23.49)
Alpha	Laboratory	25.92	23.24	21.13
		(24.43–27.41)	(21.77–24.71)	(19.84–22.43)
	Outdoor	26.14	23.67	21.26
		(24.57–27.70)	(22.08–25.25)	(20.02–22.50)
Beta	Laboratory	18.61	17.59	18.2
		(17.45–19.76)	(16.50–18.68)	(16.98–19.42)
	Outdoor	19.29	18.13	18.09
		(18.16–20.43)	(17.18–19.08)	(17.02–19.17)

Note: All values are in decibels (dB); 95% CIs in parentheses.

**Table 7 brainsci-11-00330-t007:** Analysis of variance (ANOVA) for EEG power variability.

Frequency Band	Effect	*F*(1,20)	*η* ^2^ _P_	*B* _01_	*B* _10_	*P*(*H*_0_|Data)	*P*(*H*_1_|Data)
Resting Task Theta	ENV	2.09	0.10	1.61 ^†^	0.62	0.62	0.38
	RS	10.32 **	0.34	0.06	17.22 ^†††^	0.05	0.95
	ENV × RS	0.39	0.02	3.74 ^††^	0.27	0.79	0.21
Resting Task Alpha	ENV	0.44	0.02	3.65 ^††^	0.27	0.79	0.21
	RS	30.72 ***	0.61	0.00	3823.47 ^†††^	0.00	1.00
	ENV × RS	0.04	0.00	4.49 ^††^	0.22	0.82	0.18
Resting Task Beta	ENV	4.37 *	0.18	0.57	1.74 ^†^	0.36	0.64
	RS	6.20 *	0.24	0.27	3.71 ^††^	0.21	0.79
	ENV × RS	0.10	0.01	4.34 ^††^	0.23	0.81	0.19
PASAT Theta	ENV	1.24	0.06	2.44 ^†^	0.41	0.71	0.29
	TASK	3.36	0.14	0.90	1.12 ^†^	0.47	0.53
	ENV × TASK	4.21	0.17	0.62	1.62 ^†^	0.38	0.62
PASAT Alpha	ENV	0.38	0.02	3.76 ^††^	0.27	0.79	0.21
	TASK	20.82 ***	0.51	0.00	390.76 ^†††^	0.00	1.00
	ENV × TASK	0.25	0.01	4.03 ^††^	0.25	0.8	0.2
PASAT Beta	ENV	0.61	0.03	3.34 ^††^	0.30	0.77	0.23
	TASK	0.43 **	0.02	3.67 ^††^	0.27	0.79	0.21
	ENV × TASK	2.31	0.10	1.45 ^†^	0.69	0.59	0.41

ANOVA factor labels: ENV = Environment, RS = Resting State, ELEC = Electrode, TASK = Task. *F*-value significance: * = *p* < 0.05, ** = *p* < 0.01, *** = *p* < 0.001. Bayes factor evidence strength: † = weak, †† = strong, ††† = very strong to decisive.

**Table 8 brainsci-11-00330-t008:** Observed versus Meta-Analytic Hedges’ g for EEG Power Main Effect of Resting State.

Frequency Band	Effect	Hedges’ g Observed	Hedges’ g Meta-Analysis
Theta	Laboratory	0.74	1.24
		(0.64–0.83)	
	Outdoor	0.50	
		(0.41–0.60)	(1.19–1.29)
	Collapsed	0.68	
		(0.58–0.77)	
Alpha	Laboratory	1.11	1.30
		(1.01–1.22)	
	Outdoor	0.63	
		(0.53–0.73)	(1.26–1.33)
	Collapsed	1.13	
		(1.03–1.24)	
Beta	Laboratory	0.52	0.82
		(0.42–0.61)	
	Outdoor	0.41	
		(0.32–0.50)	(0.77–0.87)
	Collapsed	0.48	
		(0.39–0.58)	

Note: 95% CIs in parentheses.

**Table 9 brainsci-11-00330-t009:** Observed versus Meta-Analytic Hedges’ g for EEG Power Main Effect of Task.

Frequency Band	Effect	Hedges’ g Observed	Hedges’ g Meta-Analysis
Theta	Laboratory	0.08	0.56
		(−0.01–0.17)	
	Outdoor	−0.36	
		(−0.45–−0.27)	(0.51–0.62)
	Collapsed	−0.21	
		(−0.30–−0.12)	
Alpha	Laboratory	−0.74	
		(−0.84–−0.64)	
	Outdoor	−0.63	−0.36
		(−0.73–−0.54)	(−0.38–−0.33)
	Collapsed	−0.79	
		(−0.88–−0.69)	
Beta-Frontal	Laboratory	0.07	0.4
		(−0.02–0.16)	
	Outdoor	−0.30	
		(−0.39–−0.21)	(0.32–0.48)
	Collapsed	−0.17	
		(−0.26–−0.08)	
Beta-Posterior	Laboratory	0.46	−0.51
		(0.36–0.55)	
	Outdoor	0.28	
		(0.19–0.38)	(−0.60–−0.42)
	Collapsed	0.39	
		(0.30–0.48)	

Note: 95% CIs in parentheses.

## Data Availability

Publicly available datasets were analyzed in this study. This data can be found at https://dataverse.tdl.org/dataverse/brainscieeg (accessed on 20 February 2020).

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
