# Peer review of "An Analysis of the External Validity of EEG Spectral Power in an Uncontrolled Outdoor Environment during Default and Complex Neurocognitive States"

_brainsci, 2021, doi:10.3390/brainsci11030330_

Round 1
Reviewer 1 Report
The study addresses an interesting research question and is, generally, well performed and clearly described. However, I noticed a number of minor issues, which should be corrected (the numbers correspond to line numbers):
15-17 "two tasks during these recordings, one engaging brain activity related to several complex cognitive functions (number sense, attention, memory, executive function) and the other maintaining a default brain state" - in fact, not two but three tasks were compared to each other (the default state was in eyes closed and eyes open variants)
63-64 consider replacing one of the two "influence(s)" with another word
68-73 a much more relevant study than Ref. 27 was published recently, where auditory oddball was paired with the same (!) activity (cycling) and the EEG was compared between two environmental conditions (quiet park vs. near a noisy roadway): Scanlon JE, Redman EX, Kuziek JW, Mathewson KE. A ride in the park: Cycling in different outdoor environments modulates the auditory evoked potentials. International Journal of Psychophysiology. Volume 151, May 2020, Pages 59-69.
130 are M and CI related to age?
136-138 were the recording in two environments done within a single experimental session (in the same day)? how quickly the recording started after participant was coming to each environment? (i.e., was there any time given for adaptation, and how much time?)
136-143 any data about the lab and outdoor environment characteristics? (e.g., temperature, noise level, any characteristics of what was within the participant's field of view, especially quantitative characteristics of light and any data about objects possibly that could attract attention, data about contrast levels and spatial frequencies, etc.)
144 was the order randomized?
146-147 "In the PASAT task, participants listened to a human voice speak a single digit once every 3 seconds"
160 "Each PASAT consisted of 60 trials"
166 "PASAT performance lasted approximately 5 – 10 minutes"
- 60 trials x 3 seconds = 3 minutes, how the total duration became longer?
174-175 was the order randomized?
175-176 any instructions about where to look? and what they could see in either environment?
219-220 was the order randomized?
228 here and below trials seems to have different meaning from what it was called trials before; consider use here and below the term "epoch" instead
247 as only two channels were actually used for the analysis, consider providing information about each of them specifically (instead of the average data about all channels)
259-260 it is not clear what "the discrepancy between the alternative hypothesis ... against the null hypothesis" means; moreover, NHST was clearly used in its literal meaning, as a "null hypothesis significance testing" (section 2.7.1 and results); probably, it should be just stated that F-values provided by ANOVA were used as an input to Bayes factors
326 is the link correct?
337 parentheses -> square brackets? (here and below)
424, 426-427 on EEG power?
532 "higher signal-to-noise ratios" -> "lower signal-to-noise ratios"?
Table 6 - something wrong with the frequency band column
Results, Discussion - it may make sense to analyze (or at least discuss) the possible effect of the environment on spectral power variance within and/or between participants, because the outdoor conditions was expected to be a less stable environment
Discussion - alpha rhythm in eye open conditions can be affected by what is in the visual field of a participant, partly because of the visual stimulation itself and partly, probably, because it can be blocked under various eye movements (e.g., Mulholland TB, Peper E. Occipital alpha and accommodative vergence, pursuit tracking, and fast eye movements. Psychophysiology. 1971 Sep;8(5):556-75), which, in turn, depends on objects, textures etc. that attract attention and provoke fixations, smooth pursuit or vergence eye movements. Visual environment can be greatly different in the lab and in the outdoor condition. Also, a participant could be somewhat less relaxed when they may expect that people not familiar with the study, including acquaintances, etc. can see them being involved into the experiment, with the EEG cap on, etc., and this may lead to lower alpha in the outdoor conditions. You may wish to discuss possible contribution of such factors.
Discussion - a non-listed limitation of the study that should be acknowledged is the limited measurement of the various characteristics of the environment that could affect the results; it is therefore not clear to which extend the results can be generalized to other environments.
Author Response
Replies to Reviewer #1
We thank the reviewer for their very helpful comments on the first version of this manuscript. We have carefully addressed the various minor issues and concerns raised by the reviewer (see below) in the revised manuscript. For ease of identifying these changes, all changes from the original manuscript are shown in red font in the revision. We have also indicated the line numbers of the changes, but please keep in mind that these line numbers are based off our submitted version of the manuscript, not the post-submission reformatted version given to the reviewer.
In addition, we must disclose that in the process of following up on these concerns, we performed an audit of our analysis pipeline and found an error in the way the linked-mastoids EEG reference was computed at the early stage of the analysis. The revised manuscript presents a full reanalysis of the data after this error was corrected. This reanalysis changed the small numerical details of the results (all in-text reporting, tables, and figures in the Results section have been updated), but not the major findings and conclusions from the original version of the manuscript except for the between-task (PASAT versus eyes open resting state) beta EEG power finding. This formerly significant finding was rendered non-significant by the corrected analysis; we re-interpret this accordingly in the revised Discussion section (see lines 648 – 667 of the revised manuscript. We apologize for this error, but are grateful for the reviewer’s suggestions that lead us to find and correct it.
Concern 1: “15-17 "two tasks during these recordings, one engaging brain activity related to several complex cognitive functions (number sense, attention, memory, executive function) and the other maintaining a default brain state" - in fact, not two but three tasks were compared to each other (the default state was in eyes closed and eyes open variants)”
Author Response: Although there were two default states utilized in this study, collectively they were performed as a single task. That said, we see the reviewers point and have modified the abstract text to reflect the engagement of two default states: “Participants performed two tasks during these recordings, one engaging brain activity related to several complex cognitive functions (number sense, attention, memory, executive function) and the other engaging two default brain states”; see lines 16 – 18 of the revised manuscript.
Concern 2: ”63-64 consider replacing one of the two "influence(s)" with another word”
Author Response: We thank the reviewer for their suggestion and have made suggested change; see lines 60 – 62 of the revised manuscript.
Concern 3: “68-73 a much more relevant study than Ref. 27 was published recently, where auditory oddball was paired with the same (!) activity (cycling) and the EEG was compared between two environmental conditions (quiet park vs. near a noisy roadway): Scanlon JE, Redman EX, Kuziek JW, Mathewson KE. A ride in the park: Cycling in different outdoor environments modulates the auditory evoked potentials. International Journal of Psychophysiology. Volume 151, May 2020, Pages 59-69.”
Author Response: We thank the reviewer for making us aware of this study, which somehow escaped out literature review. We have revised the manuscript text as follows (see lines 64 – 75 of the revised manuscript): “To our knowledge, only two mobile EEG studies to date [27,28] have directly compared human neurocognitive performance between different physical environments with different physical characteristics and levels of dynamic irregularity. The first study [27] examined differences between a controlled indoor laboratory and an uncontrolled outdoor bicycle pathway. However, cognitive task performance in this study (auditory oddball detection) was also paired with a different physical activity in each environment (sitting indoors, bicycling outside). The second study [28] removed this confound by examining cognition during bicycling activity within a quiet park and a noisy roadway. In the present study, we also recorded mobile EEG from participants performing neurocognitive tasks during the same physical activity across two different physical environments – sitting in a controlled laboratory environment (closed space, minimal noise, static temperature and atmosphere) and a moderately uncontrolled outdoor environment (open space, background noise, weather changes; see Figure 1).”
Concern 4: “130 are M and CI related to age?”
Author Response: Yes, we have edited the text here to make this clear. See line 122 of the revised manuscript.
Concern 5: “136-138 were the recording in two environments done within a single experimental session (in the same day)? how quickly the recording started after participant was coming to each environment? (i.e., was there any time given for adaptation, and how much time?)”
Author Response: Yes, the recording was performed in the two environments within a single session on the same day. How quickly the recording started in each environment depended on the order of the recordings. The recording in the first environment of the session also involved the EEG setup, which lasted approximately 30 – 40 minutes. The recording in the second environment of the session only lasted approximately 10 – 15 minutes and involved a basic setup of the recording computer, reminder of the tasks to be performed, and double-checking of the EEG setup. However, the balancing of the order of the environments across participants should have controlled for any differential adaptation effects due to this time difference. We have added text describing this in Section 2.2 Background Recording Environments (lines 127 – 167 of the revised manuscript) and Section 2.6 General Procedure (lines 248 – 255 of the revised manuscript).
Concern 6: “136-143 any data about the lab and outdoor environment characteristics? (e.g., temperature, noise level, any characteristics of what was within the participant's field of view, especially quantitative characteristics of light and any data about objects possibly that could attract attention, data about contrast levels and spatial frequencies, etc.)”
Author Response: We have added additional information about these environment characteristics in Section 2.2 Background Environments of the revised manuscript. We have also included a table reporting the average values of 6 key physical/weather variables of the outdoor environment (Temperature, Relative Humidity, Atmospheric Pressure, Wind Speed, Time of Day). This information was determined by for the date and time of each experimental session obtained from an online weather data repository (Weather Underground, www.wunderground.com). See lines 127 – 167 of the revised manuscript.
Moreover, we performed a correlational analysis to examine the relationship between these weather variables and EEG power. The basic method of analysis is described in Section 2.8.1, lines 307 – 311 and Section 2.8.2, lines 316 – 320 of the revised manuscript. The results of this analysis are reported at the end of Section 3.1 (lines 383 – 387), the end of Section 3.2 (lines 417 – 428), and the end of Section 3.3 (lines 460 – 466). We found no significant correlational relationships between these variables and EEG power.
Concern 7: “144 was the order randomized?”
Author Response: This balance was not achieved through randomization; instead an order list was created that yielded equal instances of each possible order across participants. Participants were then assigned to the next available order on the list when they arrived to the experimental session. We note, however, that due to the need to exclude some participant data due to equipment malfunction (see Section 2.1 Participants), this balancing was approximate in the final retained data set (although the deviation from the ideal balancing was small). We have added text reporting this to the revised manuscript in Section 2.2 (lines 128 – 132) and Section 2.6 (lines 240 – 247).
Concern 8: “146-147 "In the PASAT task, participants listened to a human voice speak a single digit once every 3 seconds". 160 "Each PASAT consisted of 60 trials".166 "PASAT performance lasted approximately 5 – 10 minutes" - 60 trials x 3 seconds = 3 minutes, how the total duration became longer?”
Author Response: This is a misreporting error on our part due to also including the total recorded EEG time in this statement, where the total recording time was approximately 10 minutes in order to overcome a technical recording limitation with our equipment that required us to record longer than the time it took to perform the PASAT (see Section 2.5 EEG Recording for more details on this limitation, which was also reported in the first version of this manuscript). We have revised this text as follows (see Section 2.3, lines 186 – 188) of the revised manuscript: “PASAT performance lasted approximately 3 minutes. (We note that EEG recording for the PASAT lasted this 3 minutes plus approximately another 7 minutes due to technical recording limitations; see Section 2.5 EEG Recording, below).”
In relation to this concern, we have also slightly revised the text of Section 2.5 in order to make clear the EEG recording time for each task (see lines 228 – 234 of the revised manuscript).
Concern 9: “174-175 was the order randomized?”
Author Response: For each participant, the order of eyes open/closed periods was the same in each environment, but this order was balanced between environments across participants using a predetermined order list as was done for the balancing of the order of recording environments (see Section 2.2 Background Recording Environments, above). Due to the need to exclude some participant data due to equipment malfunction (see Section 2.1 Participants), this balancing was approximately achieved in the final retained data set (although the deviation from the ideal balancing was small). See Section 2.4 (lines 196 – 198) and Section 2.6 (lines 240 – 247) of the revised manuscript.
Concern 10: “175-176 any instructions about where to look? and what they could see in either environment?”
Author Response: In the laboratory, participants were instructed to keep their eyes fixated on a small cross at the center of a computer screen, whereas in the outdoor environment participants were instructed to look straight ahead. We did not instruct the participants about what they could see in each environment. Moreover (as reported in the original manuscript and still in the revision) in both environments, participants were instructed to remain relaxed, alert, and awake, and to minimize eye movements/blinks during the recording. We have included additional detail about the resting task instruction in Section 2.4 lines 193 – 194 of the revised manuscript. We also note that we have added text in Section 2.2 Background Recording Environments about the typical field of view for the participants in the laboratory and environment; see lines 133 – 167 of the revised manuscript.
Concern 11: “219-220 was the order randomized?”
Author Response: For each participant, the order of task performance was the same in each environment, but this order was balanced between environments across participants using a predetermined order list, as was done for the balancing of the order of recording environments and for the two conditions of the resting state task (see Section 2.2 Background Recording Environments and Section 2.4 Resting State Task, above). Due to the need to exclude some participant data due to equipment malfunction (see Section 2.1 Participants), this balancing was approximately achieved in the final retained data set (although the deviation from the ideal balancing was small). See Section 2.6 lines 240 – 247 of the revised manuscript.
Concern 12: “228 here and below trials seems to have different meaning from what it was called trials before; consider use here and below the term "epoch" instead”
Author Response: We thank the reviewer for pointing this out. We have made the suggested changes in Section 2.7 of the revised manuscript.
Concern 13: “247 as only two channels were actually used for the analysis, consider providing information about each of them specifically (instead of the average data about all channels)”
Author Response: We thank the reviewer for this suggestion. We have made the suggested changes; see Section 2.7 (lines 277 – 278) of the revised manuscript.
Concern 14: “259-260 it is not clear what "the discrepancy between the alternative hypothesis ... against the null hypothesis" means; moreover, NHST was clearly used in its literal meaning, as a "null hypothesis significance testing" (section 2.7.1 and results); probably, it should be just stated that F-values provided by ANOVA were used as an input to Bayes factors”
Author Response: We thank the reviewer for this suggestion. We have revised the text as follows (see Section 2.8 lines 288 – 293): “NHST was used to compare the probability distribution of data described by an alternative hypothesis H1 of the presence of an experimental effect to the probability distribution described by the null hypothesis H0 of no effect. NHST was achieved using ANOVAs and Pearson correlation coefficients, where the associated F-values were used as an input to Bayes factors in order to compute direct probabilistic measures of the evidence for each hypothesis [54].”
If this change I unsatisfactory, we welcome any other suggested changes by the reviewer.
Concern 15: “326 is the link correct?”
Author Response: The link is correct (Section 3 line 356), but the data repository currently remains unpublished to the general public. We prefer to keep the repository unpublished until the paper reporting this data is accepted for publication. Listed below are private URLs at which the reviewer may confirm that the data is indeed uploaded to the online Texas Data Repository:
https://dataverse.tdl.org/privateurl.xhtml?token=db4fe5f5-f9e1-4888-9461-da4c5a0a23ec
https://dataverse.tdl.org/privateurl.xhtml?token=89855ff2-755d-4e01-8d01-111ec231c305
https://dataverse.tdl.org/privateurl.xhtml?token=ca745322-f826-4126-b8ba-161a1b9d1085
https://dataverse.tdl.org/privateurl.xhtml?token=f7156e7c-c5fa-48b4-8d01-2e2504311f95
https://dataverse.tdl.org/privateurl.xhtml?token=9b4b6d2c-2248-4210-ac0c-bfe62be37a34
In the event the manuscript is accepted for publication, we will immediately publish the link reported in the manuscript to give free, unconstrained access of the data to the general public.
Concern 16: “337 parentheses -> square brackets? (here and below)”
Author Response: We thank the reviewer for this suggestion. We have made the suggested changes. See Tables, 2 ,4 , 6, 8, and 9 of the revised manuscript.
Concern 17: “424, 426-427 on EEG power?”
Author Response: Yes, for EEG power. We have modified the title of these tables to make this clearer. See Tables 2 – 9 of the revised manuscript.
Concern 18: “532 "higher signal-to-noise ratios" -> "lower signal-to-noise ratios"?”
Author Response: We thank the reviewer for pointing out this typographical error. It has been corrected in the revised manuscript (Section 4.1 lines 594 – 594).
Concern 19: “Table 6 - something wrong with the frequency band column”
Author Response: This distortion in the pdf file of the manuscript appears to be an issue with the conversion of our word processing file to pdf, or is due to whatever edits were done after we have uploaded the file. We have double-checked the revised manuscript for any formatting issues that could lead to this error
Concern 20: “Results, Discussion - it may make sense to analyze (or at least discuss) the possible effect of the environment on spectral power variance within and/or between participants, because the outdoor conditions was expected to be a less stable environment”
Author Response: We have performed the requested analysis and reported it in a new section (3.3 Resting Task and PASAT EEG Power Variability Analysis) in the revised manuscript; see lines 435 – 466 and Tables 6 and 7. We used standard deviation as the variability metric because it has the same initial units as spectral power (these values were then converted to decibels. We then performed the ANOVA and Bayesian analyses as with the power data. The pattern of results from these analyses paralleled the EEG power findings. Interestingly, we found greater beta-band EEG power variability in the outdoor environment during the resting task. In the revised Discussion section (Section 4.1 lines 563 – 567), we speculate that this observation along with the resting state beta-band EEG power effects of environment suggest that it is possible that the beta-band EEG power effect reflects more variable activity of these local networks in response to the unstable outdoor environment.
Concern 21: “Discussion - alpha rhythm in eye open conditions can be affected by what is in the visual field of a participant, partly because of the visual stimulation itself and partly, probably, because it can be blocked under various eye movements (e.g., Mulholland TB, Peper E. Occipital alpha and accommodative vergence, pursuit tracking, and fast eye movements. Psychophysiology. 1971 Sep;8(5):556–575), which, in turn, depends on objects, textures etc. that attract attention and provoke fixations, smooth pursuit or vergence eye movements. Visual environment can be greatly different in the lab and in the outdoor condition. Also, a participant could be somewhat less relaxed when they may expect that people not familiar with the study, including acquaintances, etc. can see them being involved into the experiment, with the EEG cap on, etc., and this may lead to lower alpha in the outdoor conditions. You may wish to discuss possible contribution of such factors.”
Author Response: The reviewer raises good points here. However, we note that we did not observe any between-environment differences in the alpha (or theta) range for either task. This suggests that such visual factor differences between the two environments did not contribute much variance to our present measurements in the alpha or theta bands. That said, we can’t exclude the presence of such variance to some degree, so we have included a statement of this issue in the revised Discussion section (lines 538 – 553).
Concern 22: “Discussion - a non-listed limitation of the study that should be acknowledged is the limited measurement of the various characteristics of the environment that could affect the results; it is therefore not clear to which extend the results can be generalized to other environments.”
Author Response: We thank the reviewer for this suggestion. We have provided a statement of this limitation in the revised Discussion section (Section 4.3 lines 688 – 692).
Reviewer 2 Report
The article compares EGG measurements of various subjects both in and out of the laboratory. The introduction as well as the method explanation are very detailed and well justified, good work.
However, the results are very lacking. I have not found anything wrong in the analysis of the results. As mentioned no other research group has performed a similar study. This is correct, but by analyzing both environments, it is possible that in the laboratory there are more factors that aggravate the measurement than in the chosen outdoor environment. Electrical noise caused by electronic devices, power supplies and lighting may interfere with the measurement. However, in the chosen outdoor environment there are no disturbances that could be considered a factor influencing the EGG measurement.
To support this work, I consider it necessary to extend the test environments by adding disturbances such as body movements during the measurement or sources of electrical noise. This would allow the EGG to be used to analyze other types of tasks, thus broadening the scope of the low-cost EGG.
Author Response
Replies to Reviewer #2
Concern 1: “However, the results are very lacking. I have not found anything wrong in the analysis of the results. As mentioned no other research group has performed a similar study. This is correct, but by analyzing both environments, it is possible that in the laboratory there are more factors that aggravate the measurement than in the chosen outdoor environment. Electrical noise caused by electronic devices, power supplies and lighting may interfere with the measurement. However, in the chosen outdoor environment there are no disturbances that could be considered a factor influencing the EGG measurement.
To support this work, I consider it necessary to extend the test environments by adding disturbances such as body movements during the measurement or sources of electrical noise. This would allow the EGG to be used to analyze other types of tasks, thus broadening the scope of the low-cost EGG.”
Author Response: We thank the reviewer for this suggestion; however, including the influence of body movement or electrical noise is beyond the scope and goal of this study. This study was not an investigation of the efficiency of low-cost ambulatory EEG in the presence of such noise sources; there are many other studies that have investigated that particular issue (some studies of which we cite in our manuscript; see references 7, 9, 13, 15, and 16, for example).
Instead, the focus of our study was an investigation of the influence of different physical environments on neurocognitive performance as indexed via EEG power and what this says about the external validity of the specific tasks we used to index such performance. This means that we needed to minimize influences that could directly affect the EEG device. With respect to the influences mentioned by the reviewer, we removed all artifacts (including body motion artifacts) from the EEG record and examined the data within a frequency range (4 – 20 Hz) that was well below the frequency range (60 Hz) at which most electronic devices, power supplies and artificial lighting operate in our country (the USA). Thus such influences should have minimally-affected our EEG measurements in both environments.
We also note that given the focus of our study on the effects on environment on participant’s nervous system rather than the EEG device directly, it follows that there should be more disturbances in the outdoor environment that affect our measurements than in the lab. This is because the former was more complex and dynamic environment that had the potential to more greatly disturb the neurocognitive performance of our participants. That said, we take the reviewer’s point that there are many other physical variables that might affect both neurocognitive performance and performance of the EEG device. To this end, we identified 6 key weather variables (Temperature, Relative Humidity, Atmospheric Pressure, Wind Speed, Time of Day) of the outdoor environment that were either the same as in the lab or held relatively constant within the lab while fluctuating outdoors (see also our response to Concern #6, of Reviewer #1’s comments). We report the summary information about these environment characteristics in a table in Section 2.2 Background Environments of the revised manuscript. This information was determined by for the date and time of each experimental session obtained from an online weather data repository (Weather Underground, www.wunderground.com). See lines 127 – 167 of the revised manuscript. We then performed a correlational analysis to examine the relationship between these weather variables and EEG power. The basic method of analysis is described in Section 2.8.1, lines 307 – 311 and Section 2.8.2, lines 316 – 320 of the revised manuscript. The results of this analysis are reported at the end of Section 3.1 (lines 383 – 387), the end of Section 3.2 (lines 417 – 428), and the end of Section 3.3 (lines 460 – 466). We found no significant correlational relationships between these variables and EEG power.
Please note that all changes from the original manuscript are shown in red font in the revision. We have also indicated the line numbers of the changes, but please keep in mind that these line numbers are based off our submitted version of the manuscript, not the post-submission reformatted version given to the reviewer.
Finally, we also must disclose that in the process of following up on the concerns of Reviewer #1, we performed an audit of our analysis pipeline and found an error in the way the linked-mastoids EEG reference was computed at the early stage of the analysis. The revised manuscript presents a full reanalysis of the data after this error was corrected. This reanalysis changed the small numerical details of the results (all in-text reporting, tables, and figures in the Results section have been updated), but not the major findings and conclusions from the original version of the manuscript except for the between-task (PASAT versus eyes open resting state) beta EEG power finding. This formerly significant finding was rendered non-significant by the corrected analysis; we re-interpret this accordingly in the revised Discussion section (see lines 648 – 667 of the revised manuscript. We apologize for this error, but are grateful that we were able to identify and correct it.
Round 2
Reviewer 2 Report
Although a study of the different variables affecting the outdoor environment is not carried out, the conditions under which the work is carried out are recorded. The article has been modified to reflect this and is well defended against all the reviewers. Therefore, although I would liked a more in-depth study, I consider this article to be sufficient for publication.